# Overview of Multi-Modal Brain Tumor MR Image Segmentation

**DOI:** 10.3390/healthcare9081051

**Published:** 2021-08-16

**Authors:** Wenyin Zhang, Yong Wu, Bo Yang, Shunbo Hu, Liang Wu, Sahraoui Dhelimd

**Affiliations:** 1School of Information Science and Engineering, Linyi University, Linyi 276000, China; zhangwenyin@lyu.edu.cn (W.Z.); hushunbo@lyu.edu.cn (S.H.); 2Shandong Provincial Key Laboratory of Network Based Intelligent Computing, Jinan 250022, China; yangbo@ujn.edu.cn; 3School of Control Science and Engineering, Shandong University, Jinan 250061, China; wuliang@mail.sdu.edu.cn; 4School of Computer and Communication Engineering, University of Science and Technology Beijing, Beijing 100083, China; sahraoui.dhelim@xs.ustb.edu.cn

**Keywords:** image segmentation, brain tumor, magnetic resonance imaging, multi-modality

## Abstract

The precise segmentation of brain tumor images is a vital step towards accurate diagnosis and effective treatment of brain tumors. Magnetic Resonance Imaging (MRI) can generate brain images without tissue damage or skull artifacts, providing important discriminant information for clinicians in the study of brain tumors and other brain diseases. In this paper, we survey the field of brain tumor MRI images segmentation. Firstly, we present the commonly used databases. Then, we summarize multi-modal brain tumor MRI image segmentation methods, which are divided into three categories: conventional segmentation methods, segmentation methods based on classical machine learning methods, and segmentation methods based on deep learning methods. The principles, structures, advantages and disadvantages of typical algorithms in each method are summarized. Finally, we analyze the challenges, and suggest a prospect for future development trends.

## 1. Introduction

Brain tumors can grow in cerebral vessels, nerves, brain appendages and other intracranial tissues, which seriously threaten the life and health of patients. MRI plays an important role in the diagnosis and treatment of brain tumors. It is the most widely used imaging method in brain tumor detection and clinical treatment. MRI has no radiation, no injury, and no bone artifact in the human body [1]. As a multi-parameter imaging method, MRI has high resolution in soft tissue [2]. Through the acquisition of brain image detail information, we can accurately judge the pathological and histomorphological changes to optimize the segmentation results, which is helpful to the extraction of lesions and the treatment of tumors [3]. In MRI, images of different modes can be obtained according to the difference of transverse relaxation time and longitudinal relaxation time, and images of different modes have specificity in the image information. For example, T1-weighted imaging sequence (T1) can better display the anatomical structure of various brain tissues. T1-weighted Contrast-enhanced (T1C) imaging sequence can observe the boundary information of brain tumors more clearly. T2-weighted imaging sequences (T2) enhance the lesion area and are often used to identify lesions and determine tumor type. Fluid Attenuated Inversion Recovery (FLAIR) inhibited intracranial cerebrospinal fluid and was able to better detect high signal information in the lesion area [4]. In the process of diagnosis and treatment of brain tumors, accurate segmentation of brain tumor MR images is particularly important. According to different degrees of human intervention, it can be divided into artificial segmentation, semi-automatic segmentation and automatic segmentation. Traditionally artificial segmentation has high accuracy, but it is time-consuming and laborious, and subject to the subjective judgment of doctors. In addition, this method also requires experts to have both related brain tumor image knowledge and other professional knowledge of anatomy [5]. Therefore, researchers have done tremendous work on how to improve the accuracy and efficiency of brain tumor MR image segmentation by using semi-automatic segmentation and automatic segmentation. In brain images, the amount of information that can be expressed by single-mode MR images is limited, which cannot give accurate auxiliary information to doctors. The combination of different modal images can achieve complementary information between images [6] to obtain the morphological and pathological information of brain tumors. The result of the algorithm segmentation needs to be compared with the result drawn by the doctor manually, so it is at most the same as the result of the doctor’s manual segmentation. This not only saves doctors a lot of time, but also provides them with important reference information, which can assist in the diagnosis and treatment of brain tumors. Generally, the tumor segmentation process is shown in Figure 1.

This paper attempts to summarize the existing methods of brain tumor MR image segmentation. Firstly, this paper briefly introduces the database of brain tumor segmentation commonly used in brain tumor segmentation. Then, we introduce the basic ideas, network architecture, representative solutions, advantages and disadvantages of different methods. In addition, this paper compares the segmentation results of typical methods on BraTs database and clinical data. Finally, this paper analyzes the challenges faced by brain tumor MR image segmentation, and suggests prospects for development and direction.

## 2. Databases and Evaluation Measures

In the segmentation of brain tumor MR images, most research is based on public databases, and a smaller part on clinical data. After researchers obtain the results of image segmentation, they need to evaluate them. The evaluation of segmentation results can be divided into subjective evaluation and objective evaluation. Subjective evaluation needs to invest a lot of human and material resources, the evaluators rely on experience, and there is no standard answer. The subjective evaluation results of different people are generally different, and those of the same person at different times are also different. Therefore, objective evaluation measures that can be recognized by most people are particularly important in the study of brain tumor MR image segmentation.

### 2.1. Evaluation Measures Commonly Used

After continuous development and improvement, commonly used segmentation evaluation indicators are as follows: Dice Similarity Coefficient (*DSC*) [7], Jaccard Similarity (*JS*) [8], True Positive Rate (*TPR*) [9], Positive Predictive Value (*PPV*) [10] and Hausdorff Distance (*HD*) [11]. In order to obtain the evaluation measures introduced, we need to use the ground truths and actual segmentation results for calculation. The ground truth is an image formed by medical experts directly delineating the boundary of the relevant area of the lesion [12], and it is a standard that is unanimously recognized by researchers. The actual segmentation is the result of algorithm segmentation. Figure 2 shows the comparison between ground truth and the actual segmentation. 

In the Figure 2, *T*_1_ and *T*_0_ represent the tumor area and background area of the ground truth, and *P*_1_ and *P*_0_ represent the tumor region and background region of the actual segmentation result.

The value of *DSC* is between [0, 1]. When *DSC* is equal to 0, the segmentation result is the worst. On the contrary, when *DSC* is equal to 1, the segmentation result is the most accurate [7], and *DSC* [13] is computed as follows:(1)$$DSC(P_{1},T_{1})=\frac{2\left|P_{1}\cap T_{1}\right|}{\left|P_{1}\right|+\left|T_{1}\right|}$$

*JS* value is obtained by the intersection of the actual segmentation result and the ground truth and the ratio of their union, and the definition [14] is as follows:(2)$$JS(P_{1},T_{1})=\frac{\left|P_{1}\cap T_{1}\right|}{\left|P_{1}\cup T_{1}\right|}=\frac{\left|P_{1}\cap T_{1}\right|}{\left|P_{1}\right|-\left|P_{1}\cap T_{1}\right|+\left|T_{1}\right|}$$

*TPR* is obtained from the segmentation result of the algorithm and the ratio of the overlap part of ground truth to ground truth [9]. The definition of true positive rate [15] is as follows:(3)$$TPR(P_{1},T_{1})=\frac{\left|P_{1}\cap T_{1}\right|}{\left|T_{1}\right|}$$

Positive Predictive Value (*PPV*) is also called Precision. The Positive Predictive Value is obtained by the ratio of the result correctly segmented by the algorithm to the result segmented by the algorithm. The definition [10] is as follows:(4)$$PPV(P_{1},T_{1})=\frac{\left|P_{1}\cap T_{1}\right|}{\left|P_{1}\right|}$$

The definition of Hausdorff Distance (*HD*) is as follows:(5)$$HD(P_{1},T_{1})=MAX\{h(P_{1},T_{1}),h(T_{1},P_{1})\}$$
$h(A,B)=\underset{a_{{}_{i}}\in A}{max}\underset{b_{j}\in B}{min}\left\|a_{i}-b_{j}\right\|$ can be obtained from set *A* and set *B*, h(A,B) is the one-way Hausdorff distance from set *A* to set *B*, $a_{i}$ means the *i*-th point in set *A*, $b_{j}$ means the *j*-th point in set *B*, and $\left\|a_{i}-b_{j}\right\|$ means the distance between the point $a_{i}$ and $b_{j}$ [11].

### 2.2. Databases Commonly Used

The database commonly used for brain tumor segmentation is the BraTs (Brain Tumor Segmentation) database, and a small number of studies are based on clinical databases. This paper mainly introduces the BraTs2013, BraTs2015, BraTs2017, BraTs2018, BraTs2019, BraTs2020 and some clinical databases. The relevant data information involved in this paper are shown in Table 1.

From Table 1, we can see that BraTs database contains four modes: T1, T1C, T2 and Flair. All image sizes are 240 mm × 240 mm × 155 mm. Before 2018, BraTs database had training data and test data, which could be tested offline. However, since 2018, there is no test data in the database and online testing is required.

#### 2.2.1. BraTs Database

BraTs database is provided by MICCAI (Medical Image Computing and Computer Assisted Intervention) conference. This is the official database for the brain tumor MR image segmentation challenge held by the conference, and is also widely used by researchers engaged in brain tumor MR image segmentation. Since the challenge was held in 2012, the BraTs database has been updated every year. The URL of BraTs database mentioned in this paper is as follows:
BraTs2013 (Retrieved 21 May 2021 from https://www.smir.ch/BRATS/Start2013),BraTs2015 (Retrieved 21 May 2021 from https://www.smir.ch/BRATS/Start2015),BraTs2017 (Retrieved 21 May 2021 from https://www.med.upenn.edu/sbia/brats2017/data.html),BraTs2018 (Retrieved 21 May 2021 from https://www.med.upenn.edu/sbia/brats2018/data.html),BraTs2019 (Retrieved 21 May 2021 from https://www.med.upenn.edu/cbica/brats2019/data.html),BraTs2020 (Retrieved 21 May 2021 from https://www.med.upenn.edu/cbica/brats2020/data.html).


In recent years, there have been a large number of studies on BraTs series databases. Table 2 shows some of these research results.

Among the work based on BraTs2013 database, Shen et al. [19] obtained good results in the segmentation of the whole tumor and its sub-regions by using the proposed structure of one subsample and three up-samples to extract stratified features. (The network diagram is shown in Figure 3a). Zhou Z et al. [12] proposed a 3D convolution pyramid module, which is a 3D dense connection architecture that can fuse multi-scale context information. This method performs well in whole tumor segmentation.

In the work based on BraTs2015 database, Iqbal et al. [27] added jump connection and interpolation operation on the basis of Segnet (the network diagram is shown in Figure 3d), which enhanced the segmentation effect of core tumor and enhanced tumor. In the whole tumor segmentation, Casamitjana et al. [23] proposed a method that uses two paths to collect low-resolution and high-resolution features from the input image, and the segmentation effect is better than with other methods.

In the research work based on BraTs2017 database, Po et al. [26] proposed a 3D U-Net model combined with prior knowledge of lesions. Using the original image to generate a group of patients with lesion heat map, then the generated map is employed to locate the target area. (The network diagram is shown in Figure 3b). Experiments show that this method is more effective than other methods in tumor overall segmentation and enhancement segmentation.

In the research work based on BraTs2018 database, Subhashis B et al. [35] proposed a multi-plane convolution neural network for brain tumor MR image segmentation from different anatomical planes (The network diagram is shown in Figure 3e), which showed good performance in the segmentation of whole tumor, core tumor and enhanced tumor.

In the research work based on BraTs2019 database, the method of adding dense connection to encoder part of three-layer codec architecture (The network diagram is shown in Figure 3c) proposed by R Agrava et al. [44], which has the highest precision in the segmentation of whole tumor and core tumor. In the segmentation of enhanced tumor, the deep convolution neural network improved by Zhao Y et al. [43] has the best segmentation effect.

In the research work based on BraTs2020 database, the Hybrid High-resolution and Non-local Feature Network (H2NF-Net) proposed by Jia H et al. [52] uses a single and cascaded HNF-Net to segment different brain tumor regions. Combine the prediction results as the final segmentation result. (The network diagram is shown in Figure 3f). This method works well in whole tumor and core tumor segmentation tasks. In enhanced tumor segmentation, a tumor region segmentation model that combines a two-stage codec with regularization and attention mechanism proposed by Lyu C et al. [53] works well.

#### 2.2.2. Clinical Database

Clinical data of MR brain tumor images are collected by the hospital with the permission of the patients during their treatment. The collected MR brain images are used by doctors to judge the condition of patients and propose reasonable and effective treatment plans. Because of patient privacy and ethical issues, researchers are not allowed to use such data for research without permission from patients and hospitals. In recent years, the comparison of segmentation results based on clinical database is shown in Table 3.

Because the clinical data of each hospital is collected in different stages from different patients, and the equipment conditions used to collect the data are also different, it is impractical to compare the segmentation performance of these works. From the experimental results alone, the improved 3D U-Net scheme proposed by U Baid et al. [44] has higher segmentation accuracy in whole tumor, core tumor and enhanced tumor. The model consists of contraction path and expansion path. The shrinking path mainly captures the context, and the expanding path realizes the target location. The loss function, activation function and data enhancement are also considered. Therefore, each segmentation measure is increased.

## 3. Methods of Brain Tumor MR Image Segmentation

Segmentation methods of brain tumor MR image are mainly divided into three categories according to different segmentation principles: traditional segmentation methods, traditional machine learning-based segmentation methods and deep learning-based segmentation methods. Each category includes a variety of specific segmentation algorithms, as shown in Figure 4.

### 3.1. Traditional Brain Tumor Segmentation Methods

According to the different theories and emphases, the traditional segmentation methods can be generally divided into four categories: threshold based segmentation, region-based segmentation, fuzzy theory based segmentation and edge detection based segmentation [56].

#### 3.1.1. Segmentation Methods Based on Threshold

Threshold-based segmentation is the simplest method. First, it is assumed that the pixels within a range belong to the same category [57]. Brain tumor images can be divided into target region and background region by setting an appropriate threshold. Different thresholds can also be set to divide the tumor into multiple regions. After continuous research and development, the accuracy of threshold segmentation has been greatly improved. Wang Y P et al. proposed an improved threshold segmentation algorithm. The method improves the noise sensitivity in threshold segmentation by using local information of pixel neighborhood [58]. Foladivanda et al. proposed an adaptive threshold segmentation method. The method can effectively overcome the problem of uneven gray, and enhance the contrast of images, and effectively improve the DSC and JS measure of MR image segmentation of the brain tumor [59].

The segmentation method based on threshold is relatively simple, and the quality of segmentation results almost entirely depends on the size of threshold, so the selection of threshold is very important. Moreover, the threshold segmentation method can only segment simple images, and it is difficult to deal with complex images.

#### 3.1.2. Segmentation Methods Based on Region

Common region-based segmentation methods include watershed algorithm and region-growing algorithm.

Watershed algorithm is a segmentation method based on mathematical morphology. In this algorithm, the image to be processed is compared to the terrain in geography, and the elevation of terrain is represented by the gray value of the pixel. The local minimum and its adjacent area are called the ponding basin. It is assumed that there are water permeable holes at each local minimum. With the increase of infiltration water, the ponding basin will be gradually submerged. Blocking the flow of water from a stagnant basin to a nearby basin is called a dam. When the water level reaches the peak, the infiltration process ends. These dams are called watersheds. Kaleem et al. [60] proposed a watershed segmentation method guided by setting internal or external markers to calculate the morphological gradient of the input image and internal and external markers of the original image. Then they use watershed transform to obtain the segmentation results. Rajini N et al. [61] proposed a method combining threshold segmentation and watershed. First, the image was segmented by threshold method, and then the segmented image was segmented by watershed algorithm. The experiment proved that the segmentation results obtained by this method were more accurate than those obtained by one of the two methods alone, with the average TPR measure higher than 90%.

The segmentation algorithm based on watershed can obtain a complete closed curve and provide contour information for subsequent processing, whereas the watershed algorithm is influenced by noise and easy to over segment.

The region growing algorithm draws all the pixel points conforming to the criterion into the same region via formulating a criterion, so as to achieve pixel segmentation. This kind of segmentation method has the following characteristics: (1) Each pixel must be in a certain region, and the pixels in the region must be connected, and must meet certain similar conditions; (2) different regions are disjoint, and two different regions cannot have the same property. Qusay et al. [62] proposed an automatic seed region growth method, which can automatically set the initial value of seeds, avoid the defects of manual interaction, and improve the efficiency of image segmentation.

The region-based segmentation method has the characteristics of simple calculation and high accuracy, which can extract better regional features and is more suitable for segmentation of small targets. However, it is sensitive to noise and easy to make holes in the extracted region.

#### 3.1.3. Segmentation Methods Based on Fuzzy Theory

The segmentation methods based on fuzzy theory have also been highly valued. In brain tumor MR image segmentation, the most widely used Fuzzy theory algorithm is Fuzzy C-means clustering (FCM) [63]. Muneer K et al. [64] obtained the K-FCM method through the combination of FCM algorithm and K-means algorithm. The experiment proved that, compared with FCM, K-FCM showed higher accuracy in brain tumor MR image segmentation and could reduce the computational complexity. Guo Y et al. [65] proposed a Neutrosophic C-Means (NCM) algorithm based on fuzzy C-means and neutral set framework. The algorithm introduced distance constraint into the objective function to solve the problem of insufficient prior knowledge and achieved satisfactory segmentation results. On the basis of Super-pixel fuzzy clustering and the lattice Boltzmann method, Asieh et al. [66] proposed a level set method that can automatically segment brain tumors, which has strong robustness to image intensity and noise.

The segmentation method based on fuzzy theory can effectively solve the problem of incomplete image information, imprecision, and so on. It has strong compatibility and can be used in combination with other methods, but it is difficult to deal with large-scale data due to its large amount of computation and high time complexity.

#### 3.1.4. Segmentation Methods Based on Edge Detection

The segmentation principle based on edge detection and target contour achieves segmentation by obtaining the edge of the target region and then obtaining the contour of the target region. Common detection operators for edge detection include Roberts operator, Sobel operator, Canny operator and Prewitt operator [67]. Jayanthi et al. [68] integrated FCM into the active contour model. The initial contour of the model is automatically selected by FCM, which reduces the human–computer interaction. Moreover, the problem of the unclear edge contour and uneven intensity in MR images was improved. The average DSC measure of segmentation by this method reached 81%.

Compared with other traditional segmentation methods, the segmentation method based on edge detection pays attention to the edge information of the image and links the edges into contours, and the anti-noise performance is stronger. But the anti-noise performance is negatively correlated with accuracy, that is, the better the anti-noise performance, the lower the accuracy. On the contrary, improved accuracy will reduce the anti-noise performance.

### 3.2. Segmentation Methods of Brain Tumor MR Images Based on Traditional Machine Learning

Brain tumor segmentation methods based on traditional machine learning use predefined features to train the classification model. Generally, they are divided into two levels: organizational level and pixel level. At the organizational level, the classifier needs to determine which kind of organizational structure each feature belongs to, and at the pixel level the classifier needs to determine which category each pixel belongs to. Traditional Machine Learning algorithms mainly include K-Nearest Neighbors (KNN) [69], Support Vector Machine (SVM) [70], Random Forest (RF) [71], Dictionary Learning (DL) [72], etc.

Havaei et al. [69] regarded each brain as a separate database and used the KNN algorithm for segmentation. They obtained very accurate results, and the segmentation time of each brain image is only one minute, which improves the efficiency of segmentation. Llner F et al. [70] used SVM to segment brain tumors, taking into account the changing characteristics of signal intensity and other features of brain tumor MR images. The TPR measure of this method for LGG reached 83%, and the accuracy measure for HGG reached 91%. Sher et al. [73] first segmented the image by the Otsu method and K-means clustering, then extracted the features by discrete wavelet transformation, and finally reduced the feature dimension by the PCA algorithm to obtain the best features for SVM classification. The experimental results show that the sensitivity and specificity of the scheme can reach more than 90%. Vaishnavee et al. [74] used a proximal support vector machine (PSVM). The method uses equation constraints to solve the primary linear equations, which simplifies the original problem of solving convex quadratic programming. The experiment shows that PSVM is more accurate than SVM in MR image segmentation of brain tumor. Wu et al. [75] proposed a method to first segment the image into super-voxels, then segment the tumor using MRF, estimate the likelihood function at the same time, and extract the features using a multistage wavelet filter. Nabizadeh et al. [76] proposed an automatic segmentation algorithm based on texture and contour. Firstly, the initial points were determined and the machine learning classifier was trained by the initial points. Mahmood et al. [71] proposed an automatic segmentation algorithm based on RF. This algorithm uses several important features such as image intensity, gradient and entropy to generate multiple classifiers, and classifies pixels in multispectral brain MR images by combining the results to obtain segmentation results. Selvathi et al. [77] increased the weight of the wrongly classified samples and decreased the weight of the correctly classified samples in the training process. Then the classifier gives new weights to the samples to ensure that the weights of all decision trees are positively correlated with their classification ability. Finally, the input of the improved RF consists of two parts: the image intensity feature and the original image feature extracted by curve and wavelet transformation. Experimental results show that the accuracy of the improved RF scheme is 3% higher than that of the original RF algorithm. Reza et al. [78] studied the correlation of image minimization features from the perspective of image features, effectively selected features, and finally classified features in multimodal MR images through RF. Compared with the RF algorithm alone, the proposed method can improve the DSC, PPV and TPR measure simultaneously. Meier et al. [79] trained a specific random forest classifier by semi-supervised learning. It takes image segmentation as a classification task and effectively combines the preoperative and postoperative MR image information to improve the postoperative brain tumor segmentation. The PPV and ME measure obtained by this method were 93% and 2.4%, respectively. Dictionary learning is a kind of learning method for simulating dictionary lookup. The dictionary itself is set as dictionary matrix, and the method used is sparse matrix. The process of dictionary lookup is obtained by multiplying the sparse matrix and dictionary matrix, and then the dictionary matrix and sparse matrix are optimized to minimize the error between the value searched and the original data. Chen et al. [72] transformed the super-pixel feature into a high-dimensional feature space. According to the different error values of different regions when the dictionary was modeling brain tumors, the segmentation of brain tumor MR images was realized and the segmentation accuracy was improved. Li [80] proposed a multi dictionary fuzzy learning algorithm based on dictionary learning. This algorithm effectively combines dictionary learning with fuzzy algorithm, and fully considers the differences between the target region and the background, as well as the consistency within the target region. This method can describe the gray and texture information of different regions of the image, and segment the image quickly and accurately.

The traditional machine learning algorithm is better than many traditional segmentation algorithms in algorithmic performance, but there are many shortcomings when it is used in brain tumor MR image segmentation. For example, the KNN algorithm is simple to implement, and the prediction accuracy of the brain tumor region is relatively high, but the calculation is relatively large [69]. The support vector machine has strong theory, and the final result is determined by several support vectors. The calculation is relatively simple and the generalization ability is strong, but it has higher requirements concerning the selection of parameters and kernel function [70]. Random forest can solve the problem of over-segmentation well, process multiple types of data, and has good anti-noise performance. It can parallel operation and shorten the operation time, but it has a poor effect on low-dimensional tumor data processing [71]. The algorithm based on dictionary learning is similar to the idea of dimensionality reduction, both of which reduce the computing complexity and speed up the computing speed, but also have higher requirements for tumor data [72].

### 3.3. Segmentation Methods of Brain Tumor MR Images Based on Deep Learning

According to different network frameworks, the brain MR image segmentation method is based on deep learning and can be divided into that based on Convolutional Neural network (Convolutional Neural Networks, CNN) of the brain MR image segmentation method, and that based on the Convolutional Neural network (Fully Convolutional Networks, FCN) MR image segmentation method of brain tumors and the brain MR image segmentation method, based on the encoder and decoder.

#### 3.3.1. Segmentation Methods of Brain Tumor MR Images Based on CNN

Convolutional neural network belongs to the category of neural network, and its weight sharing mechanism greatly reduces the model complexity. Convolutional neural network (the network diagram is shown in Figure 5a) can directly take the image as the input, automatically extract the features, and has a high degree of invariance to the image translation, scaling and other changes. In recent years, a series of Network models based on convolutional neural Network [81], such as Network in Network [82], VGG [83], Google-Net [84], Res-Net [85], etc., have been widely used in medical image segmentation. Among them, the VGG network has a strong ability to extract features and can guarantee the convergence in the case of fewer training times. However, as the deepening of the network will cause gradient explosion and gradient disappearance, the optimization effect will start to deteriorate when the network depth exceeds a certain range.

In order to solve the problem of network degradation, He et al. [85] proposed deep Residual Network (ResNet), which achieved good results in the segmentation task [86]; Anand et al. [50] combined the 3D convolutional neural network with dense connection, pre-trained the model, and then initialized the model with the weight obtained. This method improved the DSC measure in the segmentation task of brain tumor MR images. Havaei et al. [18] constructed a cascaded dual path CNN, which took the output characteristic graph of CNN in the first stage as the additional input of CNN in the second stage. This method can effectively obtain rich background information and get better segmentation results. Lai et al. [87] reduced the tail of the original image by 98% firstly, corrected the bias field by using n4itk, then pre-segmented it by multi classification CNN, and finally obtained the final segmentation result by median filtering. The algorithm improves the DSC and PPV of segmentation significantly. Salehi et al. [6] proposed a convolutional neural network technology based on automatic context (Auto-Nets) to indirectly learn 3D image information by means of 2D convolution. This method uses 2D convolution in axial, coronal and sagittal MR images respectively to avoid complex 3D convolution operations in segmentation (The network diagram is shown in Figure 5c). Hussain et al. [88] established a correlation architecture composed of a parallel CNN layer and a linear CNN layer by adding an induction structure. This structure has achieved good results in brain tumor MR image segmentation, especially in enhancing the DSC measure to 90%. Kamnitsas et al. [24] trained 3D brain tumor images and then carried out conditional random field post-processing to obtain smoother results. Saouli et al. [89] designed a sequential CNN architecture and proposed that an end-to-end incremental network can simultaneously develop and train CNN models (the network diagram is shown in Figure 5g). The average DSC measure obtained by this method is 88%. Hu K et al. [22] proposed a more hierarchical convolution based Neural Network (Multi-Cascaded Convolutional Neural Network, MCCNN) and fully connected conditional random fields (CRFs), combined with the brain tumor segmentation method, Firstly, the brain tumor is roughly segmented by multi classification convolution neural network, and then fine segmented by fully connected random field according to the rough segmentation results, so as to achieve the effect of batch segmentation and improve the accuracy. The segmentation algorithm based on CNN can automatically extract features and process high-dimensional data, but it is easy to lose information in the process of pooling, and its interpretability is poor.

#### 3.3.2. Segmentation Methods of Brain Tumor MR Images Based on FCN

Compared with pixel-level classification, image-level classification and regression tasks are more suitable for using the CNN structure, because they both expect to obtain a probable value for image classification. For semantic segmentation of images, FCN works better. FCN has no requirement on the size of the input image, and there will be an up sampling process at the last convolution layer. This process can get the same result as the input image size, predicting each pixel while retaining the spatial information in the input image, so as to achieve the pixel classification. In simple terms, FCN is a method to classify and segment images at the pixel level. Therefore, the semantic segmentation model based on FCN is more in line with the requirements of medical image segmentation. Zhao et al. [20] proposed a combination of FCN with CRF for brain tumor segmentation. The method trains two-dimensional slices in axial, coronal and sagittal directions respectively, and then uses fusion strategy to combine segmented brain tumor images. Compared with the traditional segmentation methods, the segmentation speed is faster and the efficiency is higher. Xue et al. [90] proposed a fully convolutional neural network with feature reuse module and feature integration module (f2fcn). It reuses the features of different layers, and uses the feature integration module to eliminate the possible noise and enhance the fusion between different layers (the network diagram is shown in Figure 5e). The DSC and *PPV* obtained by this method are high. Zhou et al. [91] proposed a 3D atomic convolution feature pyramid to enhance the discrimination ability of the model, which is used to segment tumors of different sizes. Then, an improvement is made on the original basis [12], a 3D dense connection architecture is proposed, and a new feature pyramid module is designed by using 3D convolution (the network diagram is shown in Figure 5i). This module is used to fuse multi-scale context to improve the accuracy of segmentation. Liu et al. [26] proposed a Dilated Convolution optimization structure (DCR) based on Resnet-50, which can effectively extract local and global features, and this method can improve the segmentation PPV measure to 92%. The segmentation algorithm based on FCN can predict the category of each pixel, transform the image classification level to the semantic level, retain the position information in the original image, and obtain a result with the same size as the input image. However, the algorithm has low computational efficiency, takes up a lot of memory space, and the receptive field is relatively small.

#### 3.3.3. Segmentation Methods of Brain Tumor MR Images Based on Encoder-Decoder Structure

The encoder-decoder structure is generally composed of an encoder and a decoder. The encoder trains and learns the input image through a neural network to obtain its characteristic map. The function of the decoder is to mark the category of each pixel after the encoder provides the feature map, so as to achieve the segmentation effect. In the segmentation tasks based on encoder-decoder structure, the structure of encoders is generally similar, mostly derived from the network structure of classification tasks, such as VGG, etc. The purpose of doing this is to obtain the weight parameters of network training through the training of a large database. Therefore, the difference of the decoder reflects the difference of the whole network to a large extent, and is also the key factor affecting the segmentation effect.

Badrinarayanan et al. [92] proposed the SegNet model. Compared with other models, this model has a deeper layer and has better performance in semantic segmentation of pixels. The encoder part of the model consists of a 13 layer vgg-16 network, and can remember the position information of the largest pixel in the encoding phase. In the decoder, the low resolution input features are up sampled to get the segmentation results. The U-Net model based on FCN is a kind of widely used brain tumor segmentation model, in which the network structure is also made up of an encoder and a decoder, and a U-Net network jump connection will code paths, used to get the characteristics of the figure to the decoding path to the corresponding position, in order to get the characteristics of the direct sampling under the coding phase into the decoding stage, thus learning more detailed characteristics. Chen et al. [93] proposed a multi-level deep network, which can obtain image multi-level information by adding auxiliary classifiers on Multi-Level Deep Medical (MLDM) and U-Net, so as to realize image segmentation. The results of DSC, PPV and TPR were 83%, 73% and 85%, respectively. In order to reduce the semantic gap between the feature mapping of encoder and decoder networks, Zhou et al. [94] proposed a variety of nested dense connection methods to connect the encoder and decoder networks. Alom et al. [95] proposed a recursive neural network and a recursive residual convolutional neural network based on U-Net. The experimental results show that the performance of the two kinds of network segmentation combined with U-Net is better than that of U-Net alone. Zhang et al. [38] introduced the attention mechanism and residual network into the traditional U-Net network and proposed an attention residual U-Net (the network diagram is shown in Figure 5h), which improved the segmentation performance of brain tumor MR images. Milletari et al. [96] proposed the V-Net model on the basis of the 3D U-Net model, which extended the original U-Net model by using a 3D convolution check. Hua et al. [37] cascaded V-Net and used the method of segmentation of the whole tumor first into sub-regions of the tumor; the accuracy of segmentation is higher than that of direct V-Net segmentation. Cicek et al. [97] proposed a 3D U-Net model to learn the features of sparse annotated volume images. On the basis of 3D U-Net, Heet et al. [98] added a Hybrid Dilated Convolution (HDC) module to increase the sensory field of neurons, overcoming the restriction that multi-scale feature extraction requires deep neural networks. Using shallow neural networks can reduce the number of model parameters and reduce the computational complexity. Tsenget et al. [25] proposed one with the depth of the layer cross-modal convolution encoder/decoder structure, in combination with MR image data of different modalities, and at the same time using the weighted and multi-stage training methods to solve the problem of unbalanced data; compared with the traditional U-Net structure, the methods of DSC, TPR and PPV measure are improved. Isensee et al. [31] improved the U-Net network model and designed a robust neural network algorithm, which prevented overfitting by expanding the amount of data (the network diagram is shown in Figure 5f). This algorithm improved the TPR measure to 91%; Haichun et al. [28] cleverly applied the improved full convolutional neural network structure to the U-Net model and proposed a novel end-to-end brain tumor segmentation method. In this method, an up-hop connection structure was designed between the encoding path and decoding path to enhance the information flow. Jia et al. [99] constructed a HNF network based on the parallel multi-scale fusion (PMF) module, and proposed a three-dimensional high-resolution and non-local feature network (HNF-NET) for multi parameter MR imaging, which can generate strong high-resolution feature representation and aggregate multi-scale context information. The expectation maximization attention (EMA) module is introduced to extract more relevant features and reduce redundant features. The DSC and HD of the whole tumor are 91.1% and 4.13%, respectively. The segmentation algorithm based on encoder-decoder can combine high-resolution and low-resolution information, and can recognize features from multiple scales, but there is only a short connection between the encoding process and the decoding process, and the connection between the two is obviously insufficient.

### 3.4. Summary and Analysis

This paper summarizes the existing traditional machine learning based and deep learning based brain tumor MR image segmentation methods and reviews the researchers’ work in the field. It is not difficult to find that deep learning methods and techniques gradually occupy a dominant position in the field of brain tumor MR image segmentation. In the past few years, an end-to-end CNNS method and a U-Net network with codec function for brain tumor MR image segmentation have been most widely used. However, even if similar network architectures are used, the results are not identical [100,101], because data preprocessing can increase the segmentation accuracy without changing the network architecture, and can enhance the generalization ability of the network. Therefore, almost all the research has carried out data preprocessing. By comparing the segmentation performance of various methods, this paper finds that each type of method can solve some of the problems in segmentation. However, there are deficiencies in generalization. For example, brain tumor segmentation based on traditional methods is mostly simple and easy to implement, but it is difficult to process complex images, and the segmentation accuracy is generally low. Segmentation methods based on traditional machine learning are theoretically easy to understand, but it is difficult to process big data. Segmentation methods based on deep learning can extract the deep information from the image, but their interpretability is poor. The advantages and disadvantages of the brain tumor MR image segmentation method described in this paper are shown in Table 4.

In recent years, there has been more and more research into brain tumor MR image segmentation. However, the DSC measure of brain tumor segmentation is only about 0.9, which due to the complexity of the brain tumor MR image and the limitation of the segmentation algorithm. In addition, there are many other challenges in the research field of brain tumor MR image segmentation, such as the generalization ability of segmentation algorithms. Most of the existing segmentation algorithms are for a single lesion, and it is difficult to generalize these to brain tumors with different conditions or even other lesions. The proportion of brain tumor background in the MR image is too large, and the proportion of tumor target region (especially the subregion of brain tumor) is too small, so it is difficult to locate accurately and effectively in the segmentation process. MR images of brain tumors are multimodal data. If the multimodal information is not handled properly, the information between images will be confused, which can lead to no improvement, or even a reduction in segmentation accuracy. Currently, many studies on brain tumor MR image segmentation are only at the theoretical stage, unable to meet the needs of medical staff and difficult to be applied in clinical practice. Deep learning has gradually become the mainstream method in brain tumor MR image segmentation. However, as a supervised learning method, deep learning relies too much on ground truth, but manual labeling is extremely difficult.

## 4. Future Research Directions

Through studying and summarizing the existing segmentation methods, this paper looks forward to future research directions from four aspects: data acquisition and processing, feature extraction, calculation methods and clinical application.

In recent years, with the continuous development of medical imaging, MR images of brain tumors are playing an increasingly important role in the diagnosis and treatment of brain tumors. Traditional research is mostly based on the calculation and analysis of unit point and small sample data. If the data of different institutions can be integrated and utilized, the accuracy of tumor segmentation will be greatly improved [102]. However, it is still a great challenge to find a general method to deal with all changes of brain MR images from different institutions and MRI scanners. Therefore, how to make full use of multi-site and multi center data [103] will become an area worthy of attention.

Deep learning has an ability to learn features, high efficiency in extracting features, can set the number of network layers, can be mapped to any function in theory, and can solve more complex problems. As long as there are enough brain tumor MR image data, we can obtain ideal results and good portability, which can be used in Tensorflow, Pytorch and other frameworks. Therefore, deep learning based methods will continue to be active in brain tumor MR image segmentation. However, how to improve the feature expression ability of the network is the key problem in improving the performance of the segmentation network.

With the development of artificial intelligence theories and methods, there are many efficient network architectures in the field of computer vision. How to reasonably migrate these architectures to brain tumor MR image segmentation tasks, such as using mask RCNN network [104] in image retrieval and blendmark network [105] in the instance segmentation task, to improve the detection and location ability of brain tumor and its sub regions, is a direction worth exploring.

At present, the mainstream supervised brain tumor MR image segmentation methods have limited databases and are highly dependent on ground truth, while manual labeling is extremely complex. Therefore, how to segment brain tumor MR images accurately through unsupervised learning without labels, and weakly supervised learning with a small number of labels or coarse-grained labels, or to ensure that supervised methods have unsupervised learning ability, will become a hot research direction.

With the proposal of the issue of “combining scientific research with practical problems”, as well as continuous interdisciplinary collision and integration, cooperation between clinicians and computer scientists in the field of medical imaging is becoming more and more important, i.e., scientific research should meet the clinical needs of the hospital. Therefore, in the research into brain tumor MR image segmentation, how to combine clinical information, such as the deep fusion of brain tumor pathology, disease symptoms and MR image at the feature level, etc., will be an important research direction.

## Figures and Tables

**Figure 1 healthcare-09-01051-f001:**
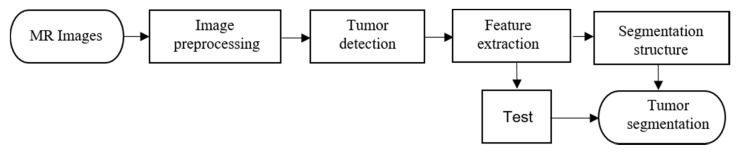
The flow chart of tumor segmentation.

**Figure 2 healthcare-09-01051-f002:**
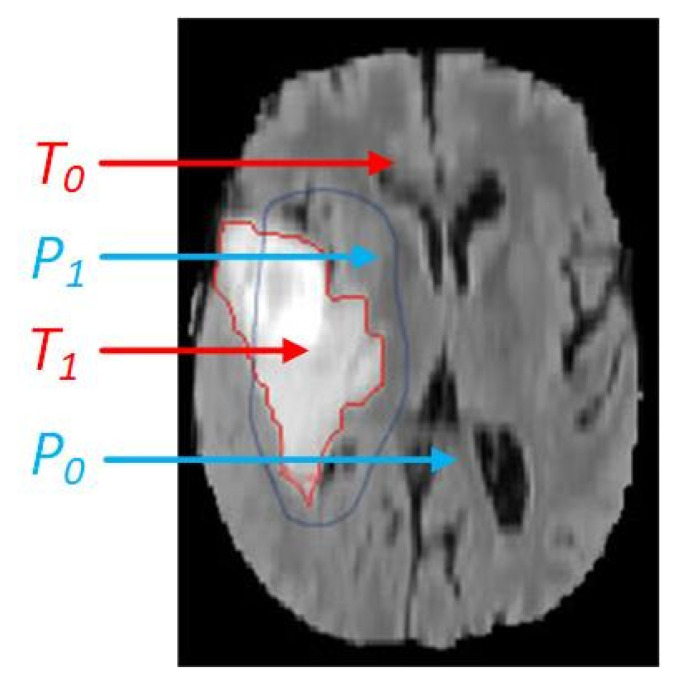
The comparison of actual segmentation and ground truth.

**Figure 3 healthcare-09-01051-f003:**
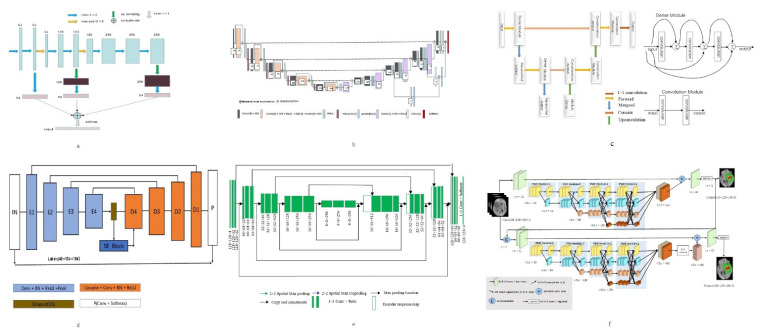
Network structure diagrams of better performing methods. (**a**) is the structure diagram of a sub sample and three up sampling structures proposed by Shen et al. [19]; (**b**) is the structure diagram of a 3D U-Net model combined with a priori knowledge of lesions proposed by Po et al. [26]; (**c**) is the structure diagram of the method of adding dense connections to the encoding part of the three tier codec architecture by R Agrava et al. [44]; (**d**) is the structure diagram of adding jump connection and interpolation operation on the basis of Segnet proposed by Iqbal et al. [27]; (**e**) is the structure diagram of a multi plane convolutional neural network proposed by Subhashis B et al. [35]; (**f**) is the structure diagram of hybrid high-resolution and nonlocal feature network (h2nf net) proposed by Jia h et al. [52].

**Figure 4 healthcare-09-01051-f004:**
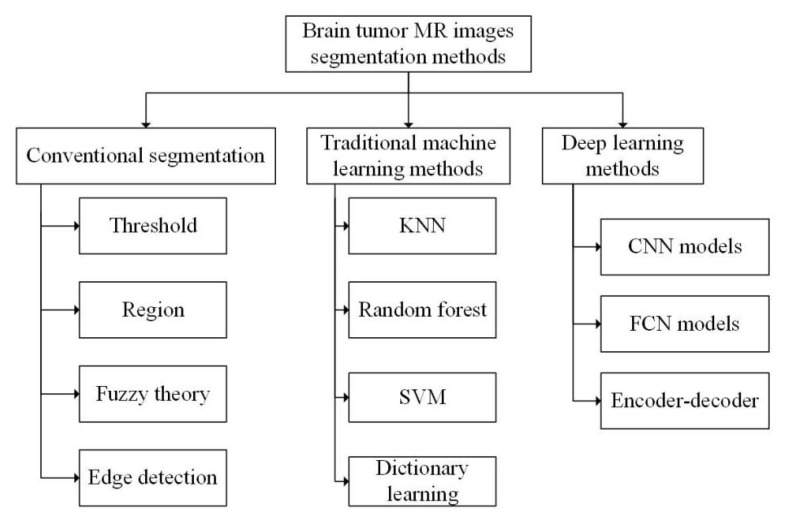
Brain tumor MR image segmentation methods.

**Figure 5 healthcare-09-01051-f005:**
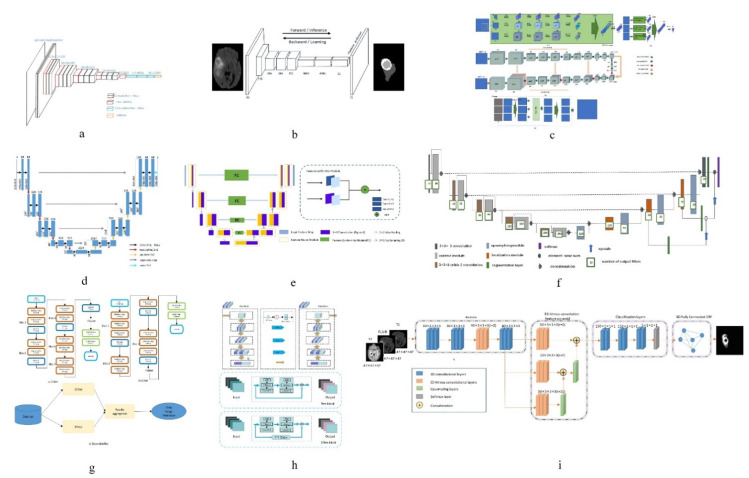
Network structure diagrams of some classical methods and improved methods. (**a**) is the classic CNN network model; (**b**) is the classic FCN network model; (**c**) is the structure diagram of a convolutional neural network technology based on automatic context (Auto-Nets) proposed by Salehi et al. [6]; (**d**) is the classic Encoder-Decoder network model; (**e**) is the structure diagram of a fully convolutional neural network with feature reuse module and feature integration module (f2fcn) proposed by Xue et al. [90]; (**f**) is the structure diagram of a robust neural network algorithm based on u-net proposed by isensee et al. [31]; (**g**) is the structure diagram of a sequential CNN architecture proposed by saouli et al. [89]; (**h**) is the structure diagram of attention residual U-net proposed by Zhang et al. [38]; (**i**) is a structural diagram of 3D dense connection combined with feature pyramid proposed by Zhou et al. [12].

**Table 1 healthcare-09-01051-t001:** Commonly used databases.

Database	Image Information	Number of Training Data	Number of Test Data	With Ground Truth		Testing Method	Data Size (mm^3^)
				Training Data	Test Data		
BraTs2013	T1, T1C, T2, FLAIR	20	10	Yes	Yes	Offline	240 × 240 × 155
BraTs2015	T1, T1C, T2, FLAIR	285	110	Yes	Yes	Offline	240 × 240 × 155
BraTs2017	T1, T1C, T2, FLAIR	285	66	Yes	Yes	Offline	240 × 240 × 155
BraTs2018	T1, T1C, T2, FLAIR	285	-	Yes	No	Online	240 × 240 × 155
BraTs2019	T1, T1C, T2, FLAIR	335	-	Yes	No	Online	240 × 240 × 155
BraTs2020	T1, T1C, T2, FLAIR	369	-	Yes	No	Online	240 × 240 × 155
Clinical database	T1, T1C, T2, FLAIR	-	-	Yes	Yes	Offline	-

**Table 2 healthcare-09-01051-t002:** Results of studies using BraTs series database in recent years.

Database	Method	Evaluation Measure: *DSC*		
		Whole Tumor	Core Tumor	Enhance Tumor
BRATS 2013	Tustison et al. [16]	0.871	0.781	0.741
	Pereira et al. [17]	0.83	0.78	0.73
	Havaei et al. [18]	0.86	0.77	0.73
	Shen et al. [19]	0.87	0.82	0.75
	Zhao et al. [20]	0.81	0.65	0.61
	P Bhagat et al. [21]	0.81	0.54	0.61
	Hu K et al. [22]	0.86	0.77	0.70
	Zhou Z et al. [12]	0.87	0.72	0.70
BRATS 2015	Casamitjana et al. [23]	0.917	0.836	0.768
	Kamnitsas et al. [24]	0.901	0.754	0.728
	Tseng et al. [25]	0.852	0.683	0.688
	Liu et al. [26]	0.87	0.62	0.68
	Iqbal et al. [27]	0.87	0.86	0.79
	Li H et al. [28]	0.890	0.733	0.726
	Hu K et al. [22]	0.87	0.76	0.75
BRATS 2017	Beers et al. [29]	0.882	0.732	0.730
	Shaikh et al. [30]	0.89	0.84	0.78
	Isensee et al. [31]	0.858	0.775	0.647
	Zhou T et al. [32]	0.885	0.846	0.734
	Po Y K et al. [33]	0.903	0.744	0.780
	Wang G et al. [34]	0.874	0.783	0.775
BRATS 2018	Wang G et al. [34]	0.908	0.869	0.807
	Subhashis B et al. [35]	0.902	0.872	0.824
	Zhou C et al. [36]	0.908	0.858	0.811
	HuA R et al. [37]	0.876	0.795	0.736
	Zhang J et al. [38]	0.876	0.810	0.773
	U Baid et al. [39]	0.848	0.769	0.668
BRATS 2019	Yogananda C et al. [40]	0.901	0.844	0.801
	Li X et al. [41]	0.886	0.813	0.771
	Wu P et al. [42]	0.891	0.817	0.757
	Zhao Y et al. [43]	0.883	0.861	0.810
	R Agravat et al. [44]	0.92	0.90	0.79
	Cheng G et al. [45]	0.905	0.820	0.764
	Ieva A et al. [46]	0.878	0.732	0.699
BRATS 2020	Lucas F et al. [47]	0.889	0.841	0.814
	Henry T et al. [48]	0.89	0.84	0.79
	Silva C et al. [49]	0.886	0.830	0.790
	Anand V et al. [50]	0.850	0.815	0.775
	Qamar S et al. [51]	0.875	0.837	0.795
	Jia H et al. [52]	0.913	0.855	0.788
	Lyu C et al. [53]	0.873	0.836	0.821

**Table 3 healthcare-09-01051-t003:** Comparison of segmentation results based on clinical databases in recent years.

Method	Data Sources	Data Volume	Evaluation Measure: *DSC*		
			Whole Tumor	Core Tumor	Enhance Tumor
Hua R et al. [37]	Local hospital	28 patients	0.864	0.804	0.722
U Baid et al. [39]	Local hospital	40 patients	0.924	0.901	0.813
Ieva A et al. [46]	Local hospital	105 patients	0.87	0.71	0.68
Shen Y et al. [54]	Local hospital	105 patients	0.894	0.790	0.653
Zhao Z et al. [55]	Local hospital	184 patients	0.785	-	-

**Table 4 healthcare-09-01051-t004:** Advantages and disadvantages of various brain tumor MR image segmentation methods.

Method		Advantage	Disadvantage
Traditional segmentation methods	Segmentation method based on threshold [57,58]	Easy to implement, Fast in calculation	Low accuracy, Meaningless for small images
	Segmentation method based on Region [61,62]	Simple calculation, High accuracy, Can operate in parallel	Sensitive to noise, Easy to produce cavity, Volume effect, Easy to oversplit
	Segmentation method based on fuzzy theory [63,66]	Low image requirements, Sensitive to parameters,	Lack of theory, Imperfect system, Long time consuming
	Segmentation method based on edge monitoring [67,68]	Strong anti noise ability, Fast detection speed	Contradiction between noise, resistance and accuracy
Segmentation method based on traditional machine learning	Segmentation method based on KNN algorithm [69]	Simple, High precision, Effective noise reduction	Data correlation required, Large amount of calculation
	Segmentation method based on random forest [71,78]	Strong fitting ability, Strong anti noise ability, Fast in calculation, balance data differences	Many features are required, Easy to lose information
	Segmentation method based on support vector machine [73,74]	Easy to fit, Strong theoretical, Easy to calculate	Sensitive to kernel function, Low precision in multitasking
	Segmentation method based on dictionary learning [72,80]	Fast operation speed, good performance	High requirements for data,
Segmentation method based on deep learning	Segmentation method based on CNN [18,50]	Shared convolution kernel, Automatic feature extraction	Weak interpretability, Easily lost information, Existence of local convergence
	Segmentation method based on FCN [26,91]	Image size is not required, Classify each pixel	Efficiency is not real-time, Insensitive to details, Lack of spatial consistency
	Segmentation method based on encoder and decoder [92,93]	Multiscale feature recognition, Combined with high and low resolution information, Restore pixel position information	Insufficient contact between encoder and decoder, A large number of parameters, Slow computing speed

## Data Availability

The URL of BraTs database mentioned in this paper is as follows: BraTs2013 (Retrieved 21 May 2021 from https://www.smir.ch/BRATS/Start2013), BraTs2015 (Retrieved 21 May 2021 from https://www.smir.ch/BRATS/Start2015), BraTs2017 (Retrieved 21 May 2021 from https://www.med.upenn.edu/sbia/brats2017/data.html), BraTs2018 (Retrieved 21 May 2021 from https://www.med.upenn.edu/sbia/brats2018/data.html), BraTs2019 (Retrieved 21 May 2021 from https://www.med.upenn.edu/cbica/brats2019/data.html), BraTs2020 (Retrieved 21 May 2021 from https://www.med.upenn.edu/cbica/brats2020/data.html).

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
