# Peer review of "Overview of Multi-Modal Brain Tumor MR Image Segmentation"

_healthcare, 2021, doi:10.3390/healthcare9081051_

Round 1
Reviewer 1 Report
The authors have taken the valiant effort of summarizing and analyzing the results of numerous brain tumor segmentation methods. The literature is expansive and the distillation of the relevant information is useful for readers.
To be a more useful resource, I believe the writing of the paper needs to be much more concise and focused. Given that potential readers of the review are already in the field on brain image segmentation, I do not think there needs to be extensive background on the imaging methods or brain tumors.
The walk-through of the Brats database results is tedious and a main takeaway from this is not clear. Personally, I would prefer that table 2-7 are merged into one full page table and the text supporting it highlight the best-performing ones.
Similar to the comment about the introductory background, I do not think an instructional section on performance metrics is needed (Section 2). The audience should be well versed in this and detracts from the focus of the paper. It also appears out of place following the database summary.
Section 3 is the heart of the paper. I think this can be improved in the machine learning and deep learning sections by highlighting fewer studies or by comparing the studies in a table format. It is very difficult to keep track of the differences in the detailed methods in the paragraph format. One paragraph extends for more than a page. I would also consider re-evaluating the figure selections for the deep learning section (fig. 4-6). These are mostly generalized reproductions (that are missing citations in the captions). I think it would be more effective to have a multi-panel figure with all the relevant network architectures drawn in the same style so that the reader may be able to visually compare. Another strategy would be to reproduce the methods figures from the best-performing studies.
In terms of conclusions, the authors do not seem to address the ceiling effect on the DSC. Even with advanced methods and larger datasets, the DSC is around 0.9. Is this comparable to human rater reliability? If not, what is the difference?
Table 9 is very important to summarize the review. It would be more readable if it were landscape and if the points were bulleted. For example, remove "it is" and just list
- sensitive to noise
- easy to produce cavity
- etc
The content of the review is valuable to the field. With a reorganization of the contents, I think it will be more accessible.
Reviewer 2 Report
In this study, authors attempted to summarize the existing methods of brain tumor MR image segmentation. This review article is useful and I recommend for publication after addressing the comments as follows
- For Table 1, With and Without in Table should be changed as Yes and No. And, authors have to provide DB’s URL that are mention in this Table.
- The performance in Table should be represented with 3 decimal if possible. Since, only 2 decimal is less information.
- Authors should provide high-quality Figures
- The highlight of this article is the information from Table 9. It is more expressive for reader if authors summarize the information in Table 9 by using only the keyword.
- In 4. Future Research Directions section, author have to discussion point by point, such as data collection, feature extraction, computational method.
- Limitation section must be provide to provide the information of the limitation of current methods
Reviewer 3 Report
In the paper 'Overview of multi-modal brain tumor MR image segmentation' by Zhang et al, the authors summarized the current methods for brain tumor MR image and segmentation. The work is important for the field but the writing needs major improvement.
1. The review paper focuses on the method of MR image and segmentation, an introduction of the work flow should be included, preferably, a chart or figure showing the general work flow would be appreciable.
2. why do the authors use all capital for the names of the authors when citing their work?
3. line 14, 'and' should be 'or'
4. Is Fig.1 and the last paragraph of Introduction really necessary? The paper should be about brain tumor MR image and segmentation methods, and I don't think the authors need to elaborate how they did the literature search.
5. In Table 1, what is the unit of data size?
6. Table 2-4, the authors might want give a little more information about the numbers listed in this table.
7. line 172-190, the font size is different from the rest of the paper.
8. In 2.1, the author listed some methods in each of the databases with any comparison or highlighting. It would be better to use one figure here to highlight one or more methods.
9. line 150-151, there seems to be some grammar errors here. The authors should check.
10. line 251-256, citations are needed here.
11. line 265, define real label and actual segmentation.
12. How is Fig. 2 related to the article? The terms mentioned in the figure were out of context.
13. line 303-305, citations are needed here.
14. line 311-313, citations are needed here.
15. Section 3, again, a figure highlighting one or more methods, e.g., show how much the segmentation improved when implementing the methods, or comparing the segmentation results of various methods should be included.
16. The writing and English language needs major improvement. The authors should consider having it polished by an English editing service.
Round 2
Reviewer 3 Report
I appreciate the authors' efforts and hard work revising the manuscript, which is now much improved. All my comments and concerns are properly addressed. The authors should still thoroughly check the manuscript for potential minor grammar errors. For example:
line 56-59: doctor's v.s. doctors'
line 136-170: work v.s. works.
Author Response
请参阅附件。
